# In situ inward epitaxial growth of bulk macroporous single crystals

Chenlong Chen[1], Shujing Sun[1], Mitch M. C. Chou[2] & Kui Xie[3]

The functionalities of porous materials could be significantly enhanced if the materials themselves were in single-crystal form, which, owing to structural coherence, would reduce electronic and optical scattering effects. However, growing macroporous single crystals remains a fundamental challenge, let alone manufacturing crystals large enough to be of practical use. Here we demonstrate a straightforward, inexpensive, versatile method for creating macroporous gallium nitride single crystals on a centimetre scale. The synthetic strategy is built upon a disruptive crystal growth mechanism that utilises direct nitridation of a parent $LiGaO_2$ single crystal rendering an inward epitaxial growth process. Strikingly, the resulting single crystals exhibit electron mobility comparable to that for bulk crystals grown by the conventional sodium flux method. This approach not only affords control of both crystal and pore size through synthetic modification, but proves generic, thus opening up the possibility of designing macroporous crystals in a wealth of other materials.

[1] Key Laboratory of Optoelectronic Materials Chemistry and Physics, Fujian Institute of Research on the Structure of Matter, Chinese Academy of Sciences, Fuzhou, Fujian 350002, China. [2] MOST Taiwan Consortium of Emergent Crystalline Materials (TECCM), Department of Materials and Optoelectronic Science, National SunYat-Sen University, Kaohsiung, Taiwan 80424, China. [3] Key Laboratory of Design & Assembly of Functional Nanostructure, Fujian Institute of Research on the Structure of Matter, Chinese Academy of Sciences, Fuzhou, Fujian 350002, China. Correspondence and requests for materials should be addressed to C.C. (email: clchen@fjirsm.ac.cn) or to M.M.C.C. (email: mitch@faculty.nsysu.edu.tw) or to K.X. (email: kxie@fjirsm.ac.cn)

Porous solid-state materials find widespread use in a variety of applications such as (electro)-catalysis, separation, photovoltaics and chemical/electrical energy storage, where the porosity affords large surface areas, boosting efficiency, capacity and reaction kinetics[1–6]. Introducing such porosity has traditionally been achieved through processing and thermal sintering of nanocrystalline powders, eventually resulting in polycrystalline ceramics, with pore sizes generally larger than 500 nm. The development of porous materials has generated much excitement and is expected to play an important role in creating new generations of (opto)-electronic devices, catalysts and supercapacitors. Synthesising porous materials is generally template based to direct the shape and size of the inorganic matrix and pores. Whereas this yields materials with regular pore size distribution and excellent long-range order of the porous structure, the inorganic matrix itself is often amorphous or at best polycrystalline. The field of modern electronics and optoelectronics, however, requires materials with extended long-range order, i.e., bulk single crystallinity to minimise losses through scattering at grain boundaries. Additionally introducing porous architecture into these single crystals would allow the development of next generation highly efficient light-emitting diodes (LED), low loss waveguides and solar cells[7]. Porosity can also be used to steer refractive indexes in photonic crystals, a highly sought after quality[8,9].

Attempts to introduce porosity into single crystals have been made through various strategies, including 'off-bubbling'[10], de-alloying[11,12], the Kirkendall effect[13], gels–aerogels transformation[14] and collective osmotic shock[15], but these often result in significant degrees of polycrystallinity or the presence of amorphous phases. Template assisted bottom-up epitaxial growth, where the template can be removed after crystal growth to leave controlled porosity, has seen much progress recently, but crystal size so far seems to be limited to the micrometre scale[6–9,16–20]. Similar issues are associated with Zn or Cd leaching during transition metal oxide nitridation[21–24] and top-down etching

approaches[25–28]. Electrochemical etching is another technique which has received attention, but its effectiveness seems confined to specific materials or architectures and is therefore not universally applicable. All progress points to a fact that synthesising bulk single crystals with porosity remains a major challenge.

GaN is a wide band gap semiconductor, whose properties are desirable in many applications, including data storage, displays and photonic devices[29–34]. Bulk single crystals on the centimetre scale have never been grown using Czochralski method, which is due to nitrogen loss and sample decomposition in molten GaN phase. Although millimetre size GaN bulk crystal can be prepared using ammonothermal methods or sodium flux method, the limitation in crystal size and the complicated preparation method hinders practical applications. In addition, only thin GaN layers can be grown onto many substrates including commercial-size $LiAlO_2$, $Al_2O_3$ and $LiGaO_2$ single crystals using hydride vapour phase epitaxy, due to the close matching of lattice parameters. However, an alternative is the nitridation of $LiGaO_2$ at elevated temperatures, causing Li and O to evaporate with concomitant formation of the GaN phase, which is then expected to prepare large-size GaN single crystal through in situ conversion of bulk $LiGaO_2$ into GaN single crystal for practical application. Moreover, $LiGaO_2$ is a suitable sacrificial substrate material since large crystals can easily be grown using the Czochralski method[35]. Previous attempts at epitaxially growing GaN by nitridation of $LiGaO_2$ have resulted in polycrystalline and amorphous films, however, and formation of an intermediate phase, '$Li_xGa_{2-x}O_{2x}N_{2(1-x)}$'[36].

Here we demonstrate in situ inward epitaxial growth of macroporous GaN bulk single crystals onto $LiGaO_2$ using chemical vapour deposition (CVD). We show that by carefully controlling the process parameters, single crystallinity is preserved with interconnected porosity over unprecedented centimetre scales. We further discuss the crucial interplay between the initial epitaxially grown thin film and subsequent crystallisation into the substrate to form the macroporous GaN single crystals.

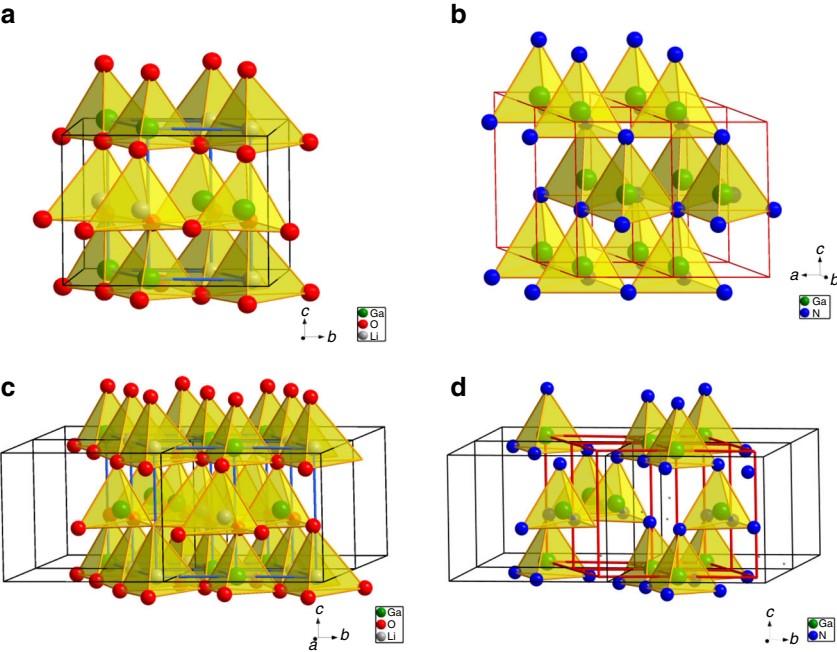

**Fig. 1** Crystal structure of $LiGaO_2$ and GaN. **a** and **c** Crystal structure of LiGaO2, **b** and **d** crystal strucure of GaN crystal strucure of GaN. Orthorhombic $LiGaO_2$ possesses a crystal structure similar to that of wurtzite GaN[50], with small lattice mismatches ($[100]_{LiGaO2}//[1\text{-}100]_{GaN} \cong 1.9\%$; $[010]_{LiGaO2}//[11\text{-}20]_{GaN} \cong -0.19\%$; $[001]_{LiGaO2}//[0001]_{GaN} \cong 3.5\%$). The Ga density in $LiGaO_2$ (2.68 g cm$^{-3}$) is approximately half that of GaN (5.12 g cm$^{-3}$), so upon epitaxially growing GaN onto a $LiGaO_2$ substrate and concomitant Li/O evaporation, half of the original lattice will remain empty, creating desired porosity

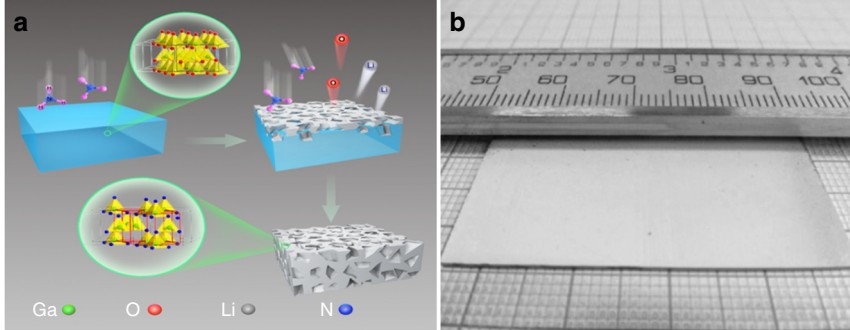

**Fig. 2** Schematics of the growth mechanism. **a** The schematic process of nitridation of LiGaO$_2$ parent single crystal to prepare macroporous GaN single crystal. **b** The prepared free-standing macroporous GaN single crystal with dimensions of 40 mm × 20 mm × 0.5 mm

Remarkably, the obtained single crystals exhibit comparable electron mobility to that of bulk crystals grown by conventional sodium flux method.

## Results

**Crystal structure**. As shown in Fig. 1, the lattices of LiGaO$_2$ and wurtzite GaN are closely related, with the former having 52% of the Ga density relative to GaN. It follows that ~48% porosity can thus be in situ introduced within the original single crystal upon nitridation and Li/O evaporation in the process of transition of LiGaO$_2$ into GaN single crystal. LiGaO$_2$ parent single crystals (001) with dimensions 40 mm × 20 mm × 0.5 mm underwent nitridation, resulting in the formation of macroporous GaN crystals with more or less the same dimensions. Figure 2a shows the schematic diagram of nitridation of LiGaO$_2$ parent single crystal to prepare macroporous single crystals. In this process, the initial nitridation produces a macroporous single-crystal film while a free-standing macroporous crystal can be anticipated after a sufficient nitridation process. And Fig. 2b is the digital photograph of a free-standing macroporous GaN single crystal with dimensions of 40 mm × 20 mm × 0.5 mm. The porosity of such a GaN crystal after full nitridation was estimated to be ~48%. To establish the nature of the crystallinity, we performed X-ray diffraction (XRD) of GaN films (partial nitridation) and bulk crystals (full nitridation). As shown in Fig. 3a, b, patterns in the 2θ/ω configurationally exhibited peaks which are characteristic of the *c*-planes for GaN and LiGaO$_2$, with the latter disappearing on extending nitridation times. The observed peaks are in line with the (001) orientation of the parent LiGaO$_2$ single crystal, indicating successful epitaxial growth of GaN, whereas the lack of diffraction peaks belonging to any other lattice planes also indicates preservation of single crystallinity upon nitridation. An XRD phi scan of (10−11) planes on a nitrided sample rotating around its *c*-axis shows peaks spaced 60° apart, confirming six-fold symmetry of these GaN planes.

**Crystal microstructure**. Field emission scanning electron microscopy (FE-SEM) was additionally used to characterise the macroporous GaN crystals. As can be seen in Fig. 3c, a cross-sectional SEM image confirms that the thickness of the fully nitrided samples is ~0.5 mm, i.e., identical to the parent LiGaO$_2$ crystal. Figure 3d also shows the resulting porosity within the GaN single crystal as epitaxially grown on the LiGaO$_2$ (001) substrates and suggests excellent three-dimensional connectivity of the pore structure. The macroporous structure of hexagonal nature covers the entire surface of the substrate material homogeneously. The typical morphology of the pores originates from the hexagonal nature of the GaN *c*-planes. In this instance, the average pore diameter is around 100 nm, but as discussed later,

pore size is controllable. The tested porosity is around 48% with an Archimedes method and this is well consistent with the calculated porosity. Although the controllable pore structure is formed with a fixed porosity, the pore/porosity could be larger if the macroporous crystal is treated using wet etching methods.

Further in-depth information about the nature of the pore structure and growth of GaN onto LiGaO$_2$ was obtained by performing transmission electron microscopy (TEM) analysis on cross-sectional samples prepared by focused ion beam (FIB). Figure 4a shows a thin film of macroporous GaN grown onto LiGaO$_2$, with good adherence and well-connected porosity. The excellent connectivity of the pore structure is further evident from TEM images of cross sections with [0002] axes, shown in the Supplementary Fig. 1, and is much improved from the one-dimension channel-like pore structures obtained through typical top-down etching methods. The fascinating 3D interconnected macroporous structure is also reflected in high-angle annular dark field (HAADF) images of the GaN [11−20] in Supplementary Fig. 2 and [1−100] zones in Supplementary Fig. 3, showing ordered isosceles triangular pores—the 2D projection of 3D regular hexagonal pyramids. A high-resolution TEM (HRTEM) image on a section of the macroporous film, Fig. 4b, clearly shows well-ordered (0001) and (11−10) lattice fringes, indicative of high crystallinity. Selected-area electron diffraction (SAED, inset in Fig. 2b) on the same section shows spots belonging exclusively to wurtzite GaN, without any other phase inclusions or signs for polycrystallinity. The LiGaO$_2$/GaN interface is characterised by good lattice correlation between both phases, as indicated in Fig. 4c. Similarly, SAED on this interface (inset in Fig. 2c) reveals spots that can only be attributed to the GaN [11−20] and LiGaO$_2$ [010] zones, reflecting the direct conversion and epitaxial relationship between substrate and newly synthesised GaN single crystal. Another HRTEM image with accompanying SAED pattern can be found in the Supplementary Fig. 4. The good agreement between the experimental convergent beam electron diffraction (CBED) pattern of the macroporous GaN crystal and a simulated pattern reveals that the top angles of the macroporous structures point to [0002] GaN[37].

**Crystal property**. To further assess the quality of the in situ grown macroporous GaN bulk single crystals, we evaluated both the optical and electrical properties by means of cathodoluminescence (CL) and Hall mobility measurements, respectively. The CL spectrum at room temperature, shown in Fig. 5a, exhibits a strong band-edge emission centred at 3.40 eV (FWHM of 0.13 eV), accompanied by negligible yellow band emission, indicating high optical quality[38–41]. The PL spectrum in Fig. 5b further confirms the high optical quality. The Hall effect, as measured at room temperature and shown in Fig. 5c, indicates an n-type macroporous GaN bulk single crystal with a resistivity of

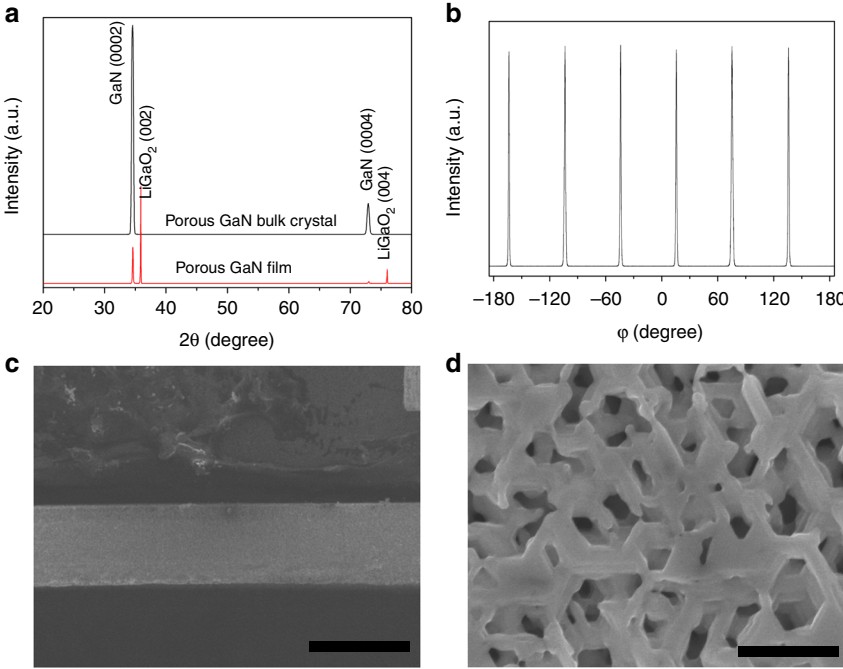

**Fig. 3** X-ray diffraction patterns and SEM images. **a** The XRD of macroporous GaN film and macroporous GaN bulk crystal with the exposed *c*-planes. **b** XRD phi scan of (10–11) planes on a nitrided sample rotating around its *c*-axis shows peaks spaced 60° apart. **c** The cross-sectional view of the free-standing macroporous GaN single crystal, and the scale bar indicates 500 μm. **d** The top view of the macroporous GaN single crystal with its exposed *c*-planes, and the scale bar indicates 200 nm

6.3 mΩ cm, electron concentration of $3.3 \times 10^{19}$ cm$^{-3}$, and Hall mobility of 30.4 cm$^2$ V$^{-1}$ s$^{-1}$. This mobility observed in our macroporous GaN samples is close to value for dense GaN bulk single crystals grown by Na flux method (30.0 cm$^2$ V$^{-1}$ s$^{-1}$)[42] and much higher than any reported polycrystalline GaN[43–46]. These properties suggest macroporous GaN has serious potential for applications in electronics and optoelectronics.

Micro-Raman was additionally used to evaluate phonon modes within the LiGaO$_2$ parent single-crystal substrate, and both macroporous GaN film and bulk single crystals as synthesised by in situ inward epitaxial growth. As detailed in Fig. 5d, after partial nitridation, phonons from both the LiGaO$_2$ substrate and GaN could be observed. The latter are those at 142.8 [E$_2$(low)], 533.5 [A$_1$(TO)], 560.7 [E$_1$(TO)], 569.4 [E$_2$(high)] and 736.8 [A$_1$(LO)] cm$^{-1}$. The appearance of A$_1$(TO) and E$_1$(TO) modes reveals the breakdown of the Raman polarisation selection rules, demonstrating the light disorder scattering as caused by the presence of an inner porous structure[39]. The signal for the strain-susceptive E$_2$ (high) phonon is shifted by ~2.4 cm$^{-1}$ towards higher frequency relative to the bulk value, indicating compressive strain of the macroporous film, due to small lattice mismatches, as well as strong chemical bonding at the interface between the macroporous GaN film and the LiGaO$_2$ (001) parent single-crystal substrate. The micro-Raman spectrum of the macroporous GaN bulk single-crystal features signals exclusively for wurtzite GaN, with phonons at 143.4 [E$_2$(low)], 533.1 [A$_1$(TO)], 559.0 [E$_1$(TO)], 567.8 [E$_2$(high)] and 738.7 [A$_1$(LO)] cm$^{-1}$. The symmetry and strength of E$_2$(high) in both films and bulk crystals confirms the high crystallinity of the macroporous GaN, whereas its position at 567.8 cm$^{-1}$ in bulk single crystals indicates a near stress-free state. These macroporous GaN in single crystal form reduces electronic and optical scattering effects and would find wide applications in the field of ideal substrate for the epitaxial growth of single crystalline GaN films, high-efficiency light-emitting diodes, hydrogen gas sensing application and excellent photoanode for photoelectrochemical cells[44–49].

## Discussion

Due to the close matching lattices, macroporous GaN single crystals can also be grown in different orientations. The *m*-plane porous GaN can be grown along the [10] direction on (100) LiGaO$_2$ and the *a*-plane porous GaN can be grown along the [11–20] direction on (010) LiGaO$_2$, respectively. XRD performed in the 2θ/ω configuration of GaN thus grown only show peaks for the GaN *m*-planes or *a*-planes, respectively, as shown in Supplementary Fig. 5, confirming single crystallinity.

Successful growth of GaN single crystals on LiGaO$_2$ is dependent on careful control of the CVD process parameters. Through systematic variation, we were able to identify the critical parameters as being temperature, system pressure and NH$_3$ flow rate. We have also established that adding hydrogen to the ammonia flow can increase the reaction kinetics, without affecting the resulting crystal quality.

At first, GaN growth is evident from temperatures as low as 750 °C, when the reaction is performed at 267 mbar and a flow rate of 200 sccm of NH$_3$, but for significant growth over 20 h, reaction temperatures of 950 °C or higher are required. Supplementary Figure 6 shows XRD patterns for samples reacted at various temperatures under the conditions described, clearly indicating the benefit of higher reaction temperatures. The FWHM of the GaN (10–10) rocking curve also shows narrowing upon increasing temperatures. SEM images in Supplementary Fig. 7 also show the typical macroporous surfaces for samples reacted at higher temperatures after 20 h, as opposed to a few surface fissures observed at 750 °C, which is further evidence of a much progressed growth of the macroporous GaN single crystals at elevated temperatures. Changing reaction temperature is also an effective method in tweaking the average pore size of the resulting macroporous single crystal. The Ga diffusion length increases with temperature, resulting in a coarser structure, with larger pore size, as shown in Fig. 4.

Secondly, the NH$_3$ flow rate is an important factor. Supplementary Figure 8 shows how the FWHM of both the GaN (0002)

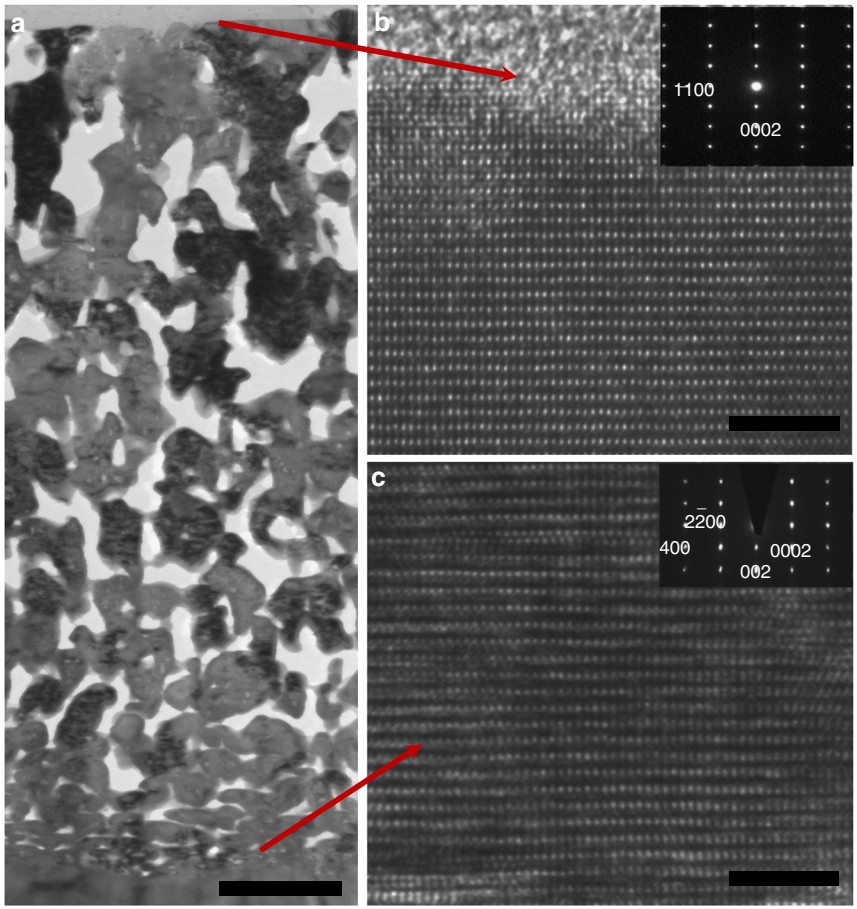

**Fig. 4** Cross-sectional TEM characterisation of a macorporous GaN film epitaxially grown on a LiGaO$_2$ (001) substrate. **a** Bright-field TEM image with [11−20] zone axis, the scale bar indicates 1 μm. **b** High-resolution TEM image of the GaN near the surface; inset: SAED pattern of the macorporous area, the scale bar indicates 5 nm. **c** High-resolution TEM image at the interface with [11−20] zone axis; inset: SAED patterns at the interface, the scale bar indicates 5 nm

and (10−10) rocking curves decrease with increasing the NH$_3$ flow rate from 50 to 600 sccm at 950 °C and a pressure of 267 mbar (20 h of reaction time), indicating an improvement of long-range order and thus crystal quality with increasing flow rate. In fact, at 50 sccm, no decent film or crystals could be grown and the sample disintegrated instead, which is indicative of poor matching of LiGaO$_2$ decomposition and GaN growth.

Thirdly, the crystal quality is additionally observed to improve with decreasing CVD system pressure, as can be seen in Supplementary Fig. 9. When increasing the pressure from 67 to 933 mbar at 950 °C and a NH$_3$ flow rate of 600 sccm, the FWHM (full width at half maximum) of the rocking curves increase after a reaction time of 20 h. At 933 mbar samples disintegrate. A careful matching of NH$_3$ flow rate and pressure is obviously required to ensure parallel decomposition of host lattice and growth of macroporous single-crystal GaN.

Finally, adding H$_2$ gas to the ammonia flow was found to improve decomposition rates for LiGaO$_2$. Similar observations have been reported previously, and the phenomenon is believed to be due to the formation of more volatile compounds, such as LiH, LiOH, LiNH$_2$ or Li$_2$NH[36,44]. Where the conversion from LiGaO$_2$ to GaN normally takes 20 h to complete at 1000 °C and 700 sccm NH$_3$ at 67 mbar, it can be completed in 1 h under identical conditions upon adding 100 sccm of H$_2$, without affecting the quality of the final GaN single crystal. Sufficient flow of NH$_3$ must be provided to ensure the growth of GaN can match the LiGaO$_2$ decomposition.

Our characterisation results confirm that 3D interconnected macroporous GaN films and bulk single crystals can be fabricated using our nitridation approach. The growth process and therefore the resulting microstructure are significantly different from previously reported porous materials, which were produced through typical etching methods, leaching or template-based procedures[1–28]. In our case, GaN is grown epitaxially on wurtzite-like LiGaO$_2$ parent single crystal under a reducing atmosphere, through decomposition of NH$_3$ into N$_2$ and H$_2$. Such a reducing atmosphere causes surface decomposition of the parent crystal, accompanied by Li and O evaporation, allowing the remaining Ga to react with N$_2$/NH$_3$ and re-crystallise as GaN. GaN initially re-crystallises epitaxially on pristine LiGaO$_2$, where the excellent lattice matching of substrate and newly formed film guarantees crystallisation with well-defined orientations (by choosing growth on a distinct face of parent single crystal). The ~52% lower crystal density of the newly formed phase leads to the formation of GaN surface islands, separated by hexagonal voids (due to hexagonal nature of GaN c-planes, Fig. 1). The spacing between islands and thus the size of hexagonal voids can effectively be controlled through the Ga diffusion length, which also determines the final pore size. To ensure continued growth of a single-crystal GaN phase, without delamination, formation of amorphous/polycrystalline phases or impurity phases, an inward crystallisation process is postulated. Such a mechanism would involve two interfaces, namely a GaN−LiGaO$_2$ one, where GaN islands form, and a LiGaO$_2$−NH$_3$ interface. The latter continually

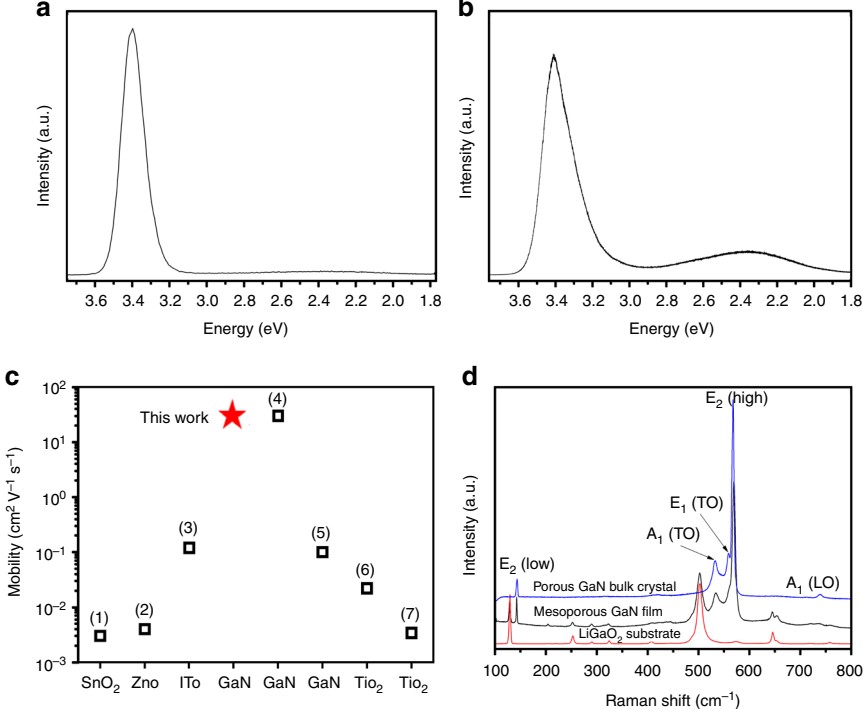

**Fig. 5** Spectroscopic characterisation and electron mobility comparison. **a** Cathodoluminescence and **b** photoluminescence spectra of the free-standing macroporous GaN single crystals at room temperatures. **c** The Hall mobility of the prepared macroporous GaN single crystal compared with other reported samples [(1), (3) and (7) mesoporous SnO₂, TiO₂ and ITO film, ref. [40]; (2) Nanoporous ZnO, ref. [39]; (4) 1-mm single crystal GaN, ref. [42]; (5) Single-crystal GaN film, ref. [45]; (6) Micrometre-size mesoporous TiO₂ single crystal, ref. [7]]. **d** The micro-Raman of LiGaO₂ parent single-crystal substrate, macroporous GaN film and bulk single crystals

decomposes, while the first creates a kinetic barrier for decomposition directly beneath the interface, preventing delamination and allowing the GaN phase to keep growing into the substrate. Still as slight mismatches between the host lattice and newly formed phase exist, this induces tensile strain (Raman spectrum, Fig. 5d) and lattice distortions in the LiGaO₂ substrate directly underneath the GaN film (HRTEM, Fig. 4c), creating a driving force for controlled decomposition of LiGaO₂ at this interface. Li/O evaporation is likely to occur laterally however, keeping the interface intact, whilst moving downwards into the substrate. This intricate interplay between lateral decomposition and inward crystallisation of GaN is believed to yield the observed three-dimensional interconnected pore structures. Such interplay is obviously subject to careful control of the CVD process parameters as discussed previously. A mismatch of decomposition rates and supply of reactant results in delamination and ultimately sample disintegration, which is also the likely cause for poor results in previous studies using a similar approach[44,45]. The N/Ga ratio (V/III) in our optimised process at 1000 °C is estimated to be 1:1, apparently providing the ideal conditions for macroporous GaN single-crystal formation.

We conduct experiments on the thermochemical properties of LiGaO₂ and GaN in NH₃ (267 mbar) at different temperatures. As shown in Supplementary Fig. 10a, c, e, the weight loss of GaN single crystal treated in ammonia for 10 h is ~0.007%, 0.06% and 0.11% at 900, 950 and 1000 °C, respectively. This means the GaN single crystal is pretty stable in ammonia even at high temperatures while the slight weight loss would be related to the Ga/N element evaporation. In contrast, the weight loss of LiGaO₂ single crystal treated in ammonia for 10 h is ~0.35%, 2.5% and 5% at 900, 950 and 1000 °C, respectively. The weight loss is related to the transformation of LiGaO₂ single crystal into GaN single crystal through an inward epitaxial growth process. This indicates that the nitridation rate of LiGaO₂ single crystal is very slow at

900 °C, while it is significantly enhanced if the temperatures are increased to 950–1000 °C, which is well consistent with the observed growth behaviour at low temperatures. As shown in Supplementary Fig. 10b, d, f, we further conduct XRD test of the nitridation process of LiGaO₂ single crystal for 10 h at different temperatures. The single crystal of GaN has formed at 900 °C while the (0002) diffraction intensity is very weak in contrast to (002) LiGaO₂ single crystal, which may be due to the macroporous GaN single crystal is pretty thin on the parent crystal surface. However, the higher nitridation temperature at 950–1000 °C facilitates the nitridation process as confirmed by the XRD diffraction.

We further consider the imperfections in the LiGaO₂ that may have influence on the growth mechanism. The LiGaO₂ crystals used in our work are grown using Czochralski growth method[35]. No cracks, domains or inclusions are observed in the bulk of our LiGaO₂ single crystals. We use chemical etching, an effective approach, to study the distribution, nature and origin of defects such as dislocations and inclusions. The defect density of the polished (001) LiGaO₂ substrates is estimated to be $4.6 \times 10^4$ cm$^{-2}$, which is quite small in large-size bulk single crystals. However, the presence of defects would favour the formation of pores in the single-crystalline GaN growth progress, while the inverse epitaxial growth process may be not largely influenced. Moreover, TEM investigation confirms that no inclusions or polycrystalline defects exist in macroporous GaN single crystal upon completion of the nitridation process of LiGaO₂ parent single crystal. Furthermore, the process can be halted at any time by removing the reactant atmosphere or by reducing the temperature. This provides flexibility and allows for the manufacture of thin films as well as bulk single crystals.

In conclusion, we have demonstrated a novel, general yet facile approach for producing large-size macroporous single crystals through in situ inward epitaxial growth on parent single crystals.

The obtained single-crystalline macroporous GaN films and bulk crystals containing 3D interconnected pores of tens to hundreds of nanometres are on an unprecedented centimetre scale, i.e., the size of a typical wafer. The pores single crystals possess good crystallinity, high optical quality and appealing electronic transport properties. The mechanically and chemically robust, biocompatible, and electrically and optically active nature of GaN single crystals, together with high surface area properties of the macroporous architecture makes it a promising building block for advanced technological applications in for instance optoelectronics and electronics. Furthermore, the CVD process used in this study is straightforward, economical and compatible with industrial GaN-based technology and would therefore be suitable for further scale-up and device integration. Moreover, our method has the potential to provide large-size macroporous GaN single crystals in large quantities at reduced cost. This novel approach in synthesising these large-scale porous single crystals should be highly adaptable as well and apply to other materials and research fields, paving the way for low-cost and high-throughput fabrication.

## Methods

**Single-crystal growth**. In situ inward epitaxial growth of single-crystalline macroporous GaN crystals: The nitridated conversion of $LiGaO_2$ crystals into macroporous GaN materials was performed in a typical CVD system. The polished $LiGaO_2$ (001) single-crystal substrate (45 mm × 20 mm × 0.5 mm) was placed in the centre of the reactor. The horizontal alumina ceramic tube reaction chamber was initially evacuated to a base pressure of $2.0 \times 10^{-3}$ torr to remove residual oxygen and then a constant flow of $NH_3$ gas (50–1000 sccm, 6 N purity) was introduced to maintain the CVD reactor at 50–700 torr. The furnace was then heated to 750–1100 °C from room temperature at a heating rate of 40 °C min$^{-1}$. The furnace temperature was maintained at 750–1100 °C for 1–72 h and then cooled naturally under the flowing gas.

**Characterisation**. The morphologies of the single-crystalline macroporous GaN samples were investigated through FE-SEM (JEOL JSM-6330TF) at an accelerating voltage of 10 kV. The orientation and structure were determined through XRD using a Bede D1 high-resolution X-ray diffractometer (UK, Bede Scientific; Cu-Kα$_1$ radiation; operated at 40 kV and 45 mA; $\lambda = 1.54052$ Å). Additional microstructural and orientational characterisation of the macroporous GaN was performed using field emission TEM (Tecnai F20 G2) at 200 kV. TEM samples were prepared using the FIB lift-out method. CL spectra were acquired using a Gatan MonoCL3 spectrometer in a JEOL JSM 6330F SEM system; these measurements were performed under an electron beam energy having an acceleration voltage of 10 kV at room temperature. Raman spectra were recorded at room temperature using a Jobin Yvon Horiba HR-800 confocal micro-Raman system coupled with a high-stability microscope. The spectrometer was calibrated using the silicon phonon line at 520.6 cm$^{-1}$. An objective of ×100 was used to collect the signal in the back scattering geometry from a focal spot (diameter: ca. 2 μm) on the samples. Raman spectra (resolution: 0.2 cm$^{-1}$) in the wave number range 100–800 cm$^{-1}$ were recorded under internal He–Ne laser excitation at a wavelength of 632.8 nm and low incident power (ca. 4 mW) to avoid sample damage or laser-induced heating.

**Data availability**. All data generated or analysed during this study are included in this published article (and its Supplementary Information files).

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

## Acknowledgements

We acknowledge Natural Science Foundation of China (61774158 and 91545123), 100 Talents Program of Fujian Province, Natural Science Foundation of Fujian Province (2016J01275) and the Ministry of Science and Technology (MOST 103-2119-M-110-003-MY3) for funding this work.

## Author contributions

C.C. conducted the experiment. M.M.C.C. and K.X. oversaw the project. C.C. and K.X. wrote the manuscript. All authors contributed to data analysis and gave approval to the final version of the manuscript.

## Additional information

**Competing interests:** The authors declare no competing financial interests.

