## [Peer Review File · Nature Communications]

Reviewers' comments:

Reviewer #1 (Remarks to the Author):

This article proposes a novel and general way for producing large-size mesoporous single crystals through in situ inward epitaxial growth on parent single crystals. The method and the result in the manuscript are interesting. I would recommend this article to be published after addressing the following issues:

1. Page 3, lines 77-78 "Li/O evaporation approximately 48% porosity can thus be introduced in situ within the original single crystal."

Page 13, lines 283-284 "The approximately 52% lower crystal density of the newly formed phase leads to the formation of GaN surface islands"

This means the porosity of the mesoporous GaN is calculated by the crystal structure of LiGaO₂ and GaN in Figure 1? The author can take method to characterize the porosity of the mesoporous GaN.

2. So, this fact means that the porosity of the mesoporous GaN fabricated by this method can not be controlled? Only the pore size is controllable in the method?

3. Can other substrates, such as Ga₂O₃ be used as parent single crystals in this in situ inward epitaxial growth method?

Reviewer #2 (Remarks to the Author):

This manuscript describes an interesting and novel topotactic crystal to porous-crystal transformation yielding single crystals with pores. The growth process is well described and analyzed. However, the claim regarding mesoporosity appears to be not supported by the data. The pores appear to be in the macropore region. If the authors would like to claim mesoporosity, porosimetry (e.g., with Ar or N₂ cryogenic adsorption) is required.

Moreover, the authors do not demonstrate if there is any potential advantage of these materials for a certain application. For these reasons I cannot recommend publication at this time.

Reviewer #3 (Remarks to the Author):

Xie presents interesting data supporting the claim of a new approach for synthesizing mesoporous, single-crystal GaN. The synthesis process is based upon the concurrent decomposition of LiGaO₂ and inward crystallization of GaN from molten Ga and decomposed NH₃. The results, specifically the structural characteristics of the GaN and its electronic and optical properties, indicate that this process is of interest to the community. However, the proposed mechanism, while plausible, is not sufficiently supported in the manuscript. The work would be significantly enhanced by providing quantitative supporting information relating the relevant thermochemical properties of LiGaO₂ and GaN to provide a process window that at least approximately matches the observations in terms of process conditions explored and experimental outcomes. The additional claim is made that this synthesis approach is generalizable to other materials systems. Such a claim cannot be supported without such quantitative support. Additional information is also needed in terms of previous attempts (referenced in the paper) and how the new approach specifically compares in terms of process and findings. Finally, the relationship between imperfections in the LiGaO₂ and impact on process could provide more information on the growth mechanism and ultimate limitations to the process. In summary, while the findings are interesting and potentially important, the lack of quantitative support of the proposed mechanism is a serious flaw in the manuscript.

Response to Reviewers' comments:

Reviewer #1 (Remarks to the Author):

This article proposes a novel and general way for producing large-size mesoporous single crystals through in situ inward epitaxial growth on parent single crystals. The method and the result in the manuscript are interesting. I would recommend this article to be published after addressing the following issues:

Answer: Many thanks for your comments. We have carefully revised this manuscript in revision.

1. Page 3, lines 77-78 “Li/O evaporation approximately 48% porosity can thus be introduced in situ within the original single crystal.” Page 13, lines 283-284 “The approximately 52% lower crystal density of the newly formed phase leads to the formation of GaN surface islands”. This means the porosity of the mesoporous GaN is calculated by the crystal structure of LiGaO₂ and GaN in Figure 1? The author can take method to characterize the porosity of the mesoporous GaN.

Answer: Thanks. The tested porosity is around 48% with an Archimedes method and this is well consistent with the calculated porosity.

2. So, this fact means that the porosity of the mesoporous GaN fabricated by this method can not be controlled? Only the pore size is controllable in the method?

Answer: Thanks very much. Yes, the controllable pore structure with a fixed porosity is formed by directly converting LiGaO₂ into GaN single crystal. The pore/porosity can be larger if the porous crystal is treated using a wet etching method.

3. Can other substrates, such as Ga₂O₃ be used as parent single crystals in this *in situ* inward epitaxial growth method?

Answer: Thanks very much. This approach has the huge potential to prepare nitride single crystal; however, the lattice match between prepared nitride and substrate is required. The conversion of single crystalline Ga₂O₃ into porous GaN single crystal would be feasible because the lattice mismatch is negligible; however, the porosity would be very small because of the similar Ga densities in Ga₂O₃ and GaN crystals.

Reviewer #2 (Remarks to the Author):

This manuscript describes an interesting and novel topotactic crystal to porous-crystal transformation yielding single crystals with pores. The growth process is well described and analyzed. However, the claim regarding mesoporosity appears to be not supported by the data. The pores appear to be in the macropore region. If the authors would like to claim mesoporosity, porosimetry (e.g., with Ar or N₂ cryogenic adsorption) is required. Moreover, the authors do not demonstrate if there is any potential advantage of these materials for a certain application. For these reasons I cannot recommend publication at this time.

Answer: Thank you very much for your comments. Yes, the pores are in the macropore region (50-100nm) in our work which has been again validated using N₂ adsorption method. We changed the word “mesoporous” into “porous” in revision.

Porous single crystals combine the porosity and single crystallinity and therefore expected to introduce new chemical and physical properties and open up new developments in materials science and applications. Porous GaN single crystals have many important and promising applications and we have added these information in revision. Porous GaN single crystal would be the ideal substrate for the epitaxial growth of single crystalline GaN films because the lattice mismatch is negligible between the film and substrate. Porous GaN single crystal is also a key materials to assemble high efficiency light-emitting diodes. Schottky diode can be developed based on porous GaN single crystal for hydrogen gas sensing application. And enhanced luminescence efficiency can be achieved if InGaN/GaN heterostructures with localized carrier are grown on porous GaN single crystal templates. Of course, porous GaN single crystal could be an excellent photoanode for photoelectrochemical cells and electron/hole separation would be ideal because of the lowest impurities in bulk.

Reviewer #3 (Remarks to the Author):

Xie presents interesting data supporting the claim of a new approach for synthesizing mesoporous, single-crystal GaN. The synthesis process is based upon the concurrent decomposition of LiGaO₂ and inward crystallization of GaN from molten Ga and decomposed NH₃. The results, specifically the structural characteristics of the GaN and its electronic and optical properties, indicate that this process is of interest to the community. However, the proposed mechanism, while plausible, is not sufficiently supported in the manuscript. The work would be significantly enhanced by providing quantitative supporting information relating the relevant thermochemical properties of LiGaO₂ and GaN to provide a process window that at least approximately matches the observations in terms of process conditions explored and experimental outcomes. The additional claim is made that this synthesis approach is generalizable to other materials systems. Such a claim cannot be supported without such quantitative support. Additional information is also needed in terms of previous attempts (referenced in the paper) and how the new approach specifically compares in terms of process and findings. Finally, the relationship between imperfections in the LiGaO₂ and impact on process could provide more information on the growth mechanism and ultimate limitations to the process. In summary, while the findings are interesting and potentially important, the lack of quantitative support of the proposed mechanism is a serious flaw in the manuscript.

Answer: Many thanks for your comments. We have carefully revised this manuscript and we have also conducted supplementary experiments to further support the mechanism.

In this work, we have demonstrated the growth of porous GaN single crystal through an inward epitaxial method and the lattice match is ideal between GaN and LiGaO₂ single crystals. This nitridation would be a potential approach to transform other parent single crystal into a porous nitride single crystal if the lattice mismatch is acceptable between each other. In revision, we have corrected the “general approach” into “potential approach”.

Previous attempts at epitaxially growing GaN by nitridation of LiGaO₂ have resulted in polycrystalline and amorphous films however and formation of an intermediate phase, ‘Li_xGa_{2-x}O_{2x}N_{2(1-x)}’. Normally these attempts have tried high ammonia pressure and low ammonia flow rates; however, these conditions are not suitable for GaN single crystal growth but favour the decomposition of LiGaO₂ single crystals. In our work, we show that by carefully controlling the process parameters, single crystallinity of GaN is preserved with interconnected porosity over unprecedented centimetre scales. Through systematic variation, we are able to identify the critical parameters as being

temperature, system pressure and NH₃ flow rate. We have also established that adding hydrogen to the ammonia flow can increase the reaction kinetics, without affecting the resulting crystal quality.

In order to support the growth mechanism, we also conducted supplementary experiments on the thermochemical properties of LiGaO₂ and GaN in NH₃ at different temperatures. As shown in Fig.1(a), (c) and (e), the weight loss of GaN single crystal treated in ammonia (267 mbar) for 10 hours are approximately 0.007%, 0.06% and 0.11% at 900, 950 and 1000°C, respectively. This means the GaN single crystal is pretty stable in ammonia even at high temperatures while the slight weight loss would be related to the element evaporation. In contrast, the weight loss of LiGaO₂ single crystal treated in ammonia (267 mbar) for 10 hours are approximately 0.35%, 2.5% and 5% at 900, 950 and 1000°C, respectively. The weight loss is related to the transformation of LiGaO₂ single crystal into GaN single crystal through an inward epitaxial growth process. This indicates that the nitridation rate of LiGaO₂ single crystal is very slow at 900°C while it is significantly enhanced if the temperatures are increased to 950-1000°C. As shown in Fig.1(b), (d) and (f), we conducted *ex situ* XRD the nitridation process of LiGaO₂ single crystal for 10 hours at different temperatures. The single crystal of GaN has formed at 900°C while the (0002) diffraction intensity is very weak in contrast to (002) LiGaO₂ single crystal, which may be due to the porous GaN single crystal is pretty thin on the parent crystal surface. However, the higher nitridation temperature at 950-1000°C facilitates the nitridation process as confirmed by the XRD diffraction. We have carefully revised the growth mechanism in revision.

Fig.1 The weight loss of LiGaO₂ and GaN single crystal in NH₃ (267 mbar) at different temperatures for 10 hours: (a) 900°C, (c) 950 °C and (e) 1000°C; The *ex situ* XRD of LiGaO₂ single crystal after nitridation in ammonia (267 mbar) for 10 hours: (b) 900°C, (d) 950 °C and (f) 1000°C.

The main defects in LiGaO₂ crystals are inclusions, dislocations and interfaces. The inclusions are γ -Ga₂O₃ which are approximately in nano-scale. The straight-line dislocations are of edge type and lie in the (001) plane Fig.2(a). The image width of dislocation was broadened owing to the segregation of inclusions, as indicated by arrow A. Arrow B indicates inclusions in nanoscale. It can be partly solved by using off-stoichiometry initial charge with excess of Li₂O. Interfaces parallel to (001) prevail in LiGaO₂ pulled along (100) as shown in Fig.2(b) and (c). The domain formation is most possibly attributed to the polarization inversion. [K. Xu, et al., Journal of Crystal Growth 216 (2000) 343].

Fig.2. (a) Edge dislocations in LiGaO₂ crystals; (b) and (c) X ray topography of LiGaO₂ (010) slices, $g=[002]$.

The LiGaO₂ crystals used in our work are grown by ourselves, as shown in Fig.3. No cracks, domains or inclusions were observed in our LiGaO₂ bulk. Chemical etching is a very effective technique for studying the distribution, nature and origin of defects such as dislocations and inclusions. In our work, the chemical etching combined with SEM test demonstrate that the density of nanoscale defect is estimated to be $4.6 \times 10^4/\text{cm}^2$ in the polished (001) LiGaO₂ substrates, which is quite small in large-size bulk single crystals. [C. Chen, C. Li, S. Yu and M. Chou, Journal of Crystal Growth 402 (2014) 325.] The presence of defects would favor the formation of pores in the single crystalline GaN growth progress while the inverse epitaxial growth process may be not influenced.

Fig.3 (a) Bulk LiGaO₂ single crystal; (b) Polished (001) LiGaO₂ substrates; (c) Etching pits morphology of dislocation emerged on the substrate.

REVIEWERS' COMMENTS:

Reviewer #1 (Remarks to the Author):

The authors have addressed all my questions in a satisfactory manner. I have no further criticisms and feel this is a very interesting study.

Reviewer #2 (Remarks to the Author):

The authors have addressed my comments. I suggest that the term macroporous is used throughout the manuscript including the title. It is important to communicate the pore size range achieved unambiguously.

Response to Reviewers' comments:

Reviewer #1 (Remarks to the Author):

The authors have addressed all my questions in a satisfactory manner. I have no further criticisms and feel this is a very interesting study.

Answer: Many thanks for your comments. We have again carefully revised this manuscript.

Reviewer #2 (Remarks to the Author):

The authors have addressed my comments. I suggest that the term macroporous is used throughout the manuscript including the title. It is important to communicate the pore size range achieved unambiguously.

Answer: Many thanks. We have used the "macroporous" in our manuscript including the title.